# Host-Associated Distribution of Two Novel Mammarenaviruses in Rodents from Southern Africa

**DOI:** 10.3390/v15010099

**Published:** 2022-12-29

**Authors:** Marike Geldenhuys, Jacqueline Weyer, Teresa Kearney, Wanda Markotter

**Affiliations:** 1Centre for Viral Zoonoses, Department of Medical Virology, Faculty of Health Sciences, University of Pretoria, Pretoria 0001, South Africa; 2Centre for Emerging Zoonotic and Parasitic Diseases, National Institute for Communicable Diseases of the National Health Laboratory Services, Johannesburg 2131, South Africa; 3Department of Microbiology and Infectious Diseases, School of Pathology, University of Witwatersrand, Johannesburg 2131, South Africa; 4Ditsong National Museum of Natural History, Pretoria 0001, South Africa; 5Department of Zoology and Entomology, University of Pretoria, Pretoria 0001, South Africa

**Keywords:** mammarenavirus, Mariental virus, Lunk virus, Miseq amplicon sequencing, surveillance

## Abstract

Mammarenaviruses are hosted by several rodent species, a small number of which have been known to be zoonotic. Host surveillance among small mammals has identified a large diversity of previously undescribed mammarenaviruses. Intensified biosurveillance is warranted to better understand the diversity of these agents. Longitudinal host surveillance involving non-volant small mammals at a site in the Limpopo province, South Africa, was conducted. The study reports on the screening results of 563 samples for the presence of mammarenavirus RNA. PCR-positive samples were subjected to sequencing using Miseq amplicon sequencing. Sequences with close similarity to Mariental and Lunk viruses were identified from two rodent species, *Micaelamys namaquensis* and *Mus minutoides.* This represents the first description of these viruses from South Africa. The genomic sequences reported here partially satisfied the requirements put forward by the International Committee on the Taxonomy of Viruses’ criteria for species delineation, suggesting that these may be new strains of existing species. The known distribution of these mammarenaviruses is thus expanded further south in Africa.

## 1. Introduction

Rodents (Rodentia) are a large order of mammals that are important hosts for a number of zoonotic viral agents, including mammarenaviruses. According to the International Committee on the Taxonomy of viruses (ICTV) taxonomy, the genus *Mammarenavirus,* family *Arenaviridae,* comprises 39 species of 46 viruses. These species can be further grouped into the New World and Old World complexes based on antigenic and phylogenetic similarities and geographic distribution [1]. Rodents are often persistently infected with mammarenaviruses, and human exposures occur during contact with contaminated rodent excreta (urine, fecal material, and saliva) that arise opportunistically due to the peri-domestic habits of many of these hosts [2,3]. Lassa fever exerts a significant zoonotic disease burden in Western African countries in which it is endemic due to the host range of its reservoir *Mastomys* sp. [4]. Many Lassa virus (LASV) infections are mild or even asymptomatic, though severe clinical cases have an observed fatality rate of 15% [5]. Another emergent arenavirus, Lujo virus (LUJV), exemplifies the need for active viral surveillance to discover and describe this diversity. LUJV was responsible for a small outbreak cluster with extremely high fatalities in 2008, when a critically ill person was airlifted from Zambia to a hospital in South Africa [6]. Both the index case and three persons in direct contact with the patient died of hemorrhagic fever, with a fifth person recovering. Since the first outbreak, no additional cases of LUJV have been reported, and the virus’s host origins remain unknown.

Biosurveillance studies in several African rodent species have identified a large diversity of mammarenaviruses (see [7]), such as Mobala (MOBV), Ippy, Mopeia (MOPV), Gairo (GAIV), Luna (LUNV), Lunk and Mariental viruses [8,9,10,11,12,13,14,15,16]. These Old-World arenaviruses have mostly originated from genera within the *Muridae* family (e.g., *Mastomys*, *Praomys*, *Mus*, *Grammomys*, and *Micaelamys*) from a wide geographical distribution (West, Central, East, and Southern Africa). Moreover, since mammarenaviruses are also generally associated with specific rodent species, with speciation possibly affected by host-switching, recombination, and reassortment [7], investigating related species to identify the diversity of potentially zoonotic agents among non-volant small mammals is of increasing interest. The zoonotic potential of many mammarenaviruses has also not been determined, though serological findings suggest that exposures to widespread viruses such as MOPV from Southern Africa have occurred [8].

Current species demarcation criteria by the International committee on the taxonomy of viruses (ICTV) for arenaviruses include both genetic and phenotypic features [17]. Viral species are differentiated if there is less than 80% and 76% nucleotide sequence identity within the S and L genomic segments, respectively. An amino acid divergence greater than 12% within the nucleoprotein gene represents another criterion to delimit separate species. Additional features include the association with distinct hosts (or sympatric hosts), distinct geographical distribution, and the ability to cause human disease.

Here, we report the findings of mammarenavirus seasonal host surveillance carried out over two years among non-volant small mammals (rodents, sengis, and shrews) in the Limpopo province of South Africa. The finding suggests widespread distribution of mammarenaviruses closely associated with Mariental and Lunk mammarenaviruses in Southern Africa.

## 2. Materials and Methods

### 2.1. Study Site

The study was a collaboration between the Biosurveillance and Ecology of Emerging Zoonoses research group at the University of Pretoria, the National Institute for Communicable Diseases, and the Ditsong National Museum of Natural History. The Meletse region (24.5914° S, 27.6258° E) in the Rooiberg area near Thabazimbi in Limpopo, South Africa is a rich ecological site (Figure 1). Various species of non-volant mammals, including rodents, sengis, and shrews, inhabit the region [18]. The site of interest includes the Madimatle cave (inhabited by bats such as *Miniopterus* and *Rhinolophus* spp.) [19] and surrounding natural landscape that is grazed by both domestic and wildlife animals. The Madimatle cave is also frequented by local cultures for religious purposes and ancestor worship.

### 2.2. Collection of Samples

Samples were collected following biosafety protocols informed by risk assessment and included the use of personalized protective equipment such as double layer nitrile examination gloves, thick leather gloves for animal handling, disposable surgical gowns (Foliodress Hartmann, Johannesburg, South Africa) over coveralls (Tyvek, DuPont, Johannesburg, South Africa), gumboots and powered air-purifying respirators (3M, Johannesburg, South Africa). All equipment, PPE, and sample containers were decontaminated after use with a 10% liquid bleach solution (5500 ppm hypochlorite solution), and disposable PPE was discarded as biohazardous waste.

Sampling activities were performed as five trips to the Meletse region throughout the year, ensuring sampling across all seasons. Surveillance began in February 2015 and ended in November 2017, totaling 15 sampling sessions. Small non-volant mammals were caught with the use of Sherman traps (H.B Sherman Traps, Inc., Tallahassee, FL, USA) and snap traps across seven trap locations, which were baited with a ca. 10 mm bait ball (a mixture of peanut butter and rolled oats) placed on the trigger board inside the trap. The traps were opened two hours before sunset and checked at sunrise the next morning. Captured animals were transferred to cotton-holding bags until sample collection. Voucher specimens were collected from approximately three males and three females per species at each site per trip from February–November (excluding July, October, and December). The age and sex of each mammal were recorded before sampling, and animals not taken as vouchers were released at the capture site.

As part of a larger biosurveillance study, several samples were collected, including fur, excreta (urine and feces), ectoparasites, oral swabs, and blood [20]. Only fecal material and kidney tissues were utilized in this study. Sterile forceps were used to collect fecal pellets directly from mammals or cotton-holding bags. Following non-destructive sample collection, all animals were anesthetized using isoflurane (Safeline Pharmaceuticals, Johannesburg, South Africa) for safe and humane collection of blood from the lateral saphenous vein/lateral tail vein, ventral tail artery (1–3% vol/body mass). Individuals that were released also received a unique tattoo on the tail for mark-recapture. Tattoos were placed on the tail after cleaning with AIMS animal tissue prep (AIMS™) by using round liner (7RL) needles in a commercial tattoo gun (Bold Monk Ordinary Tattoo Machine, South Africa) at 18 V on the AC converted analogue power supply (GetInked, Pretoria, South Africa). Animals taken as voucher specimens received an isoflurane overdose for euthanasia. During necropsy, using sterile scissors and forceps, the following tissues were collected: brain, tongue, salivary glands (where possible), pectoral muscles (for barcoding), heart, lungs, kidneys, spleen, liver, reproductive organs, intestine, rectum, bladder, oral swabs, faeces (when available) and fur. Collected tissue material was inactivated in 1x DNA/RNA Shield (ZymoResearch, Irvine, CA, USA), and animal carcasses were inactivated in 10% formalin for 24 h and transferred to ethanol for longer storage. Voucher specimens were submitted to the Ditsong National Museum of Natural History’s Small Mammal collection and the Natural History Collection of Public Health and Economics. All samples were stored dry in 2 ml cryogenic storage tubes and immediately frozen in a dryshipper (MVE Vapor Shippers, Ball Ground, GA, USA). Samples were extracted in the BSL3 facility at the Department of Medical Virology, Centre of Viral Zoonoses of the University of Pretoria, after which all molecular testing was performed in a BSL2 laboratory.

### 2.3. Morphological and Molecular Host Identification

Identifications in the field of captured mammals and later extracted skulls of voucher specimens were made using morphological characteristics and measurements [21,22,23] (as indicated in Appendix A). Genetic barcoding was performed on select specimens to associate with taxonomic identification based on morphology. Partial Cytochrome B sequences were produced as reported by Greenberg et al. [24]. Briefly, 5 µL of extracted DNA was added to a 50 µL reaction volume containing 25 mM magnesium chloride, 1× Dream Taq buffer (Thermo Scientific, Johannesburg, South Africa ), 1.25 U Dream Taq (Thermo Scientific, USA), 10 mM dNTPs (Thermo Scientific, Johannesburg, South Africa), 20 pmol each of CytB For (Appendix A) and CytB Rev (Integrated DNA technologies, Coralville, IA, USA), and 28.75 µL nuclease free water (Ambion, Thermo Scientific, Johannesburg, South Africa). Cycling conditions were as follows: 94 °C for 2 min; 45 cycles of 94 °C for 30 s, 55 °C for 45 s and 72 °C for 1 min 30 s; and 72 °C for 10 min. Amplicons of correct size as viewed on a 1.5% agarose gel (Lonza, Johannesburg, South Africa) were excised and purified with the Zymoclean™ Gel DNA Recovery Kit (ZymoResearch, Irvine, CA, USA). Purified amplicons were prepared for sequencing with the BigDye Terminator v3.1 Cycle Sequencing Kit (Thermo Fisher Scientific, USA) according to manufacturer’s recommendations for 10 μL reactions and purified with the ethanol/EDTA/sodium acetate precipitation method. Sequencing was performed on the ABI 3500xl at the DNA Sequencing facility of the University of Pretoria. Cytochrome B barcodes were searched on National Center for Biotechnology Information’s (NCBI) BLAST function to identify similar host species and confirm morphological identifications.

### 2.4. Molecular Surveillance for Mammarenaviruses

Kidney and fecal material were immersed in 1x DNA/RNA Shield (ZymoResearch, USA) and homogenized using two 5 mm sterile stainless-steel beads (Qiagen, Hilden, Germany) in the TissueLyser II system (Qiagen, Hilden, Germany) for 45 s at 30 Hz. Nucleic acids were extracted from homogenates with the ZR-Duet DNA/RNA MiniPrep Plus kit (ZymoResearch, Irvine, CA, USA) to facilitate the extraction of both nucleic acids for viral surveillance and host barcoding. Complementary DNA was prepared as 20 μL randomly primed reactions using 100 ng random primers (Integrated DNA Technologies, Coralville, IA, USA) and Superscript IV (Thermo Scientific, Johannesburg, South Africa) according to manufacturer’s recommendations. RNA remaining in cDNA was treated with 5 U RNase H (Thermo Scientific, Johannesburg, South Africa) and 2 μL RNase H buffer (Thermo Scientific, Johannesburg, South Africa), followed by incubation for 20 min at 37 °C and inactivation at 65 °C for 10 min. Surveillance for mammarenavirus nucleic acids was performed using a modified assay by Vieth et al. [25] which targets a 395 bp region of the L segment. Randomly primed cDNA was used as the template for a 50 µL Dream Taq PCR assay consisting of the following: 1.25 U Dream Taq polymerase and 1× Dream Taq buffer (Thermo Scientific, Johannesburg, South Africa), 10 mM dNTP mix (Thermo Scientific, Johannesburg, South Africa), 2 μL randomly primed cDNA and 10 pmol each of the primers LVL3359A-plus, LVL3359D, LVL3359G-plus, LVL3754A-minus, LVL3754D-minus (Appendix A) as well as 36.75 μL nuclease-free water (Ambion, Thermo Scientific, Johannesburg, South Africa). All primers were synthesized by Integrated DNA technologies. The cycling conditions were 95 °C for 2 min, followed by 45 cycles of 95 °C for 20 s, 55 °C for 30 s, and 72 °C for 1 min, with a final extension of 72 °C for 10 min. All positive amplicons of the correct sizes were purified and sequenced as described above. Tissue tropism was investigated only for virus-positive individuals, where all available sample material was extracted as described and tested for the presence of mammarenaviruses. Hantavirus surveillance was also conducted according to Klempa [26], though no nucleic acids were identified (see Appendix A).

### 2.5. Genome Characterization of Mammarenaviruses

The genomes of detected arenaviruses were recovered through amplicon-based Illumina sequencing on the MiSeq platform. To reduce secondary structure interference in subsequent PCRs cDNA with Superscript IV (Thermo Scientific, Johannesburg, South Africa) was again made as before, though with adjusted cycling conditions (50 °C for 30 min, 55 °C for 5 min, 50 °C for 20 min, 60 °C for 1 min, and 50 °C for 10 min; inactivated at 70 °C for 15 min) [27,28]. RNA was removed by RNase H treatment as described before. The complete S segment was amplified with primers targeting the ends of the segment [27]. Briefly, 5 μL cDNA, 1 U Phusion High-Fidelity DNA polymerase, 1X Phusion GC Buffer (New England BioLabs, Ipswich, MA, USA), 10 mM dNTPs (Thermo Scientific, USA), forward (LV-SJ 1-plus) and reverse primers (LV-SJ 3402-minus) was combined with 3% DMSO and 27 μL nuclease free water (Ambion, Thermo Scientific, Johannesburg, South Africa) for a 50 uL reaction. Cycling conditions were as follows: 98 °C for 20 s, 35 cycles each of 98 °C for 20 s, 63 °C for 30 s, 72 °C for 90 s, followed by a 10 min hold at 72 °C. The longer L segment was amplified in 2 amplicons using primers (Appendix A) [25,28], namely LVL1-plus and LVL3754A-minus, as well as LVL3359A-plus with LVL7279-minus. The assays were similarly performed with Phusion High-Fidelity DNA polymerase as described before, with the addition of 3 mM magnesium chloride (Thermo Scientific, Johannesburg, South Africa) for the second overlapping L RT-PCR. Reaction conditions were as follows: 98 °C for 30 s, 35 cycles each of 98 °C for 20 s, 61 °C for 30 s, 72 °C for 90 s, followed by a 10 min hold at 72 °C. Amplified products were gel purified and concentration determined with Qubit High Sensitivity double-stranded DNA assay (Thermo Scientific, Johannesburg, South Africa). Amplicon regions were combined in equal proportions per segment per virus requiring four libraries. Sequencing was done at the National Institute for Communicable Diseases Sequencing Core Facility (Sandringham, South Africa). After library prep with the Nextera XT Library prep kit (Illumina, San Diego, CA, USA), the amplicons were sequenced at 10 million reads. CLC Genomics Workbench v6 was used to quality trim raw reads, and paired-end reads were used to perform de novo assemblies to recover full segments. NCBI’s ORF finder (online tool available at https://www.ncbi.nlm.nih.gov/orffinder/, accessed on 23 January 2022) was utilized to validate open reading frames assembled for correct genes and functional proteins. Sequences were submitted to Genbank with accession numbers: OL790924-OL790927.

Following segment assembly with de novo assembly, the sequences toward the 5′ and 3′ ends of the segments and regions of significant difference in comparison to reference genomes were confirmed by re-amplification with sequence-specific primers (using kidney and lung RNA extracts) and Sanger sequencing. The primer sequences and binding locations are listed in Appendix A.

### 2.6. Phylogenetic Analysis and Genome Comparisons

All nucleotide and amino acid sequence alignments were performed with CLUSTALW in the BioEdit sequence alignment editor (v.7.2.5) [29]. Relevant reference genomes of described species and sequences from BLAST analyses of sequences from this study were collected from GenBank (NCBI). Estimations of pairwise similarities were performed with p-distance analyses in MEGA v.7 [30] and the PAirwise Sequence Comparison (PASC) program from NCBI. NCBI’s Open Reading Frame Finder was used to estimate ORF’s and enable complete genome annotation. Phylogenetic analyses of all segments were performed with maximum likelihood in MEGA v7 and Bayesian phylogenetics using BEAST v.1.10.4 [31]. CIPRES Science Gateway was used to run jModelTest2 and BEAST. Maximum likelihood trees were constructed with Jukes Cantor model for nucleic acids and the LG model with gamma distribution for amino acids, with 1000 bootstrap replicates. Bayesian maximum clade credibility trees were constructed with the general time reversible (GTR) model with gamma distribution and invariant sites. Bayesian MCMC chains were set to 10,000,000 states, sampling every 1000 steps, and convergence was confirmed via an effective sample size (ESS) of >200. Final Bayesian trees were calculated in Tree Annotator with a 10% burn-in. Trees were viewed and edited in Figtree v.1.4.2. Glycoprotein annotations were inferred from gene comparisons and motif descriptions from literature [32,33,34].

## 3. Results

### Biosurveillance for Mammarenaviruses

A total of 318 fecal and 245 kidney samples were collected from small non-volant mammals in the Meletse region between November 2015 and November 2017 (Table 1). The samples were collected from a total of 425 animals, for which both fecal and kidney material were available from 138 animals, with only kidney tissue from 107 animals and only fecal for another 180 animals. Most mammals caught were from the genera *Aethomys*, *Gerbilliscus*, and *Mastomys* (>100 individuals). Sample sizes per species per month varied due to the opportunistic nature of the trapping. Samples collected from shrews were limited to only 14. Arenavirus RNA was identified from rodent samples, specifically, two kidney samples and one fecal sample, derived from UP12472 *Mus minutoides* and UP12291 *Micaelamys namaquensis.* Further tropism investigation identified arenavirus RNA in the liver, lung, and spleen of UP12472 (though not from fecal material) and in the urine, fecal, lung, liver, and spleen of UP12291.

Based on the sequence information from the 395 bp region of the L segment, the sequence from UP12291 was most related to Mariental mammarenavirus strain N27 MRMi.n9 (NC_027136.1) collected in Namibia in 2012 [11]. Likewise, the sequence obtained for UP12472 was most closely related to Lunk mammarenavirus NKS-1 that was identified from Zambia in 2010 (NC_018711.1) [10]. To improve the assessment of the genetic relationship of these sequences, complete genomes were produced using amplicon-based sequencing on the MiSeq platform. The organization of the sequenced genome segments is indicated in Figure 2.

The complete segments and entire open reading frames (ORFs) for each detected arenavirus were compared to reference species Mariental (NC_027134 and NC_027136) and Lunk (NC_018710 and NC_018711) mammarenaviruses (Table 2). The L segment of Mariental virus and UP12291/M. namaquensis/LP/RSA/2017 differed by 34.4% and was largely due to a variable region in the L gene ORF of approximately 870 nt. This same region was highlighted by Tešíková et al. [9] as a possible artefact from chimeric assembly with the original Mariental virus S segment. A BLAST search of this variable 870 bp region from the Mariental L segment confirmed its similarity to the S segment. The corresponding region within UP12291/M. namaquensis/LP/RSA/2017 was both confirmed with Sanger sequencing as accurately assembled and yielded close matches with L segments of other arenaviruses. This variable region was trimmed for a more accurate similarity between Mariental and UP12291/M. namaquensis/LP/RSA/2017 in the L segments as 74% and L gene alone as 73.9% nucleotide identity (75.7% amino acid similarity).

Comparisons between Lunk virus and UP12472/M. minutoides/LP/RSA/2017 identified high percentage similarities of the S segment genes (NP and GP), though differences between the lengths of the L gene on the L segment. The L gene ORF of Lunk virus is only 6246 nt in length (2081 aa) compared to the 6645 nt (2214 aa) of UP12472/M. minutoides/LP/RSA/2017. Other arenaviruses such as Lymphocytic choriomeningitis mammarenavirus (NC_004291) and Souris mammarenavirus (NC_039011) also possess L genes of over 6600 nt in length (6632nt/2210aa and 6647 nt/2215aa, respectively). Alignments of the L gene segments indicate that the L gene ORF of the available Lunk virus starts at nucleotide 394 compared to other viruses, corresponding to 131 amino acids ‘absent’ from the start of the ORF. Inspection of the gene alignments indicates a missing nucleotide at position 78 of Lunk virus in relation to UP12472/M. minutoides/LP/RSA/2017 and Souris virus L gene ORF’s, altering the translational frame and resulting in stop codons. The L gene ORF of Lunk virus thus starts only downstream at position 394 (in comparison to other arenavirus genomes). Whether this represents a miss-assembly following the sequencing of Lunk virus or a natural deletion within its genome is uncertain. The L gene ORF of Lunk virus and UP12472/M. minutoides/LP/RSA/2017 thus share ~72% nt similarity (76.4–77.3% aa).

Phylogenies of the structural genes and whole segments were constructed to determine the closest ancestors of UP12291/M. namaquensis/LP/RSA/2017 and UP12472/M. minutoides/LP/RSA/2017. For the NP gene, the gene was trimmed to approximately 912 nt to include a larger number of the newly described diversity (Figure 3). Phylogenies of the complete S segment, L gene and glycoprotein (Appendix A) are in agreement with the topology of the nucleoprotein gene (Figure 3).

The closest branches to UP12291/M. namaquensis/LP/RSA/2017 (and Mariental virus from Namibia) include Bobomene, Solwezi and Ippy viruses, and share 77–86.5% amino acid similarities within the 304 amino acid region (912-nucleotides) of the nucleoprotein. UP12472/M. minutoides/LP/RSA/2017 (and Lunk virus from Zambia) share the closest similarities to Rat arenavirus 1/YN2013/CHN, Lymphocytic choriomeningitis virus (LCMV), and Dandenong viruses with 83.7–85.3% amino acid identity within this nucleoprotein region. The results confirm that the identified arenaviruses are not identical to recently reported arenaviruses from South African rodent surveillance [8] and instead are most related to arenaviruses present in members of the same host species previously reported from other African countries (Namibia and Zambia) (see PASC plots as Appendix A for comparisons).

Annotations of the glycoprotein open reading frame (Appendix A) depict similar conserved features as other mammarenaviruses. The GP1 subunit of the glycoprotein contributes to interactions with the host receptor and shares 88.6% amino acid identity between UP12472/M. minutoides/LP/RSA/2017 and Lunk virus, and 79.3% identity between UP12291/M. namaquensis/LP/RSA/2017 and Mariental virus. Several Old World arenaviruses, such as LCMV, LASV, MOPV, and MOBV viruses are known to utilize the receptor alpha-dystroglycan (specifically the matriglycan carbohydrate within this receptor) [35,36]. Comparisons of the Lunk-lineage viruses to LCMV GP1 subunit identify an amino acid similarity of 66–68.6% (that drops to below 23% when compared to LASV, MOPV). Comparable similarities are seen when comparing the Mariental-lineage viruses with LASV and MOPV of 61.2–64.8% (and below 24.5% when compared to LCMV). The terminal amino acids of the GP1 region contain the conserved ‘RRLL’ recognition motif and are essential for cleavage of the glycoprotein subunits by a signal peptidase (SPase) and subtilisin kexin isozyme-1/site-1 (SKI-1) protease [36]. Variation in this motif can be seen within the newly described viruses, as ‘RRLM’ for UP12472/M. minutoides/LP/RSA/2017 (with similar residues seen for certain MOPV variants [36] and among LUNV and MOBV viruses; Appendix A) and ‘RRIL’ for UP12291/M. namaquensis/LP/RSA/2017. The presence of Ile261 instead of Leu261 within this motif appears novel to the Mariental lineage (in comparison to other Old World mammarenaviruses Appendix A).

Host species from which arenaviruses were detected were confirmed with both morphological measurements (Natural History Collection of Public Health and Economics voucher numbers NHCPHEMAM-1 and NHCPHEMAM-2, Appendix A) and genetic cytochrome B barcoding comparisons (sequences were submitted to Genbank with accession numbers: OP785691 and OP785692). UP212291 cytochrome B sequence match *M. namaquensis* sequences from South Africa with 99.1% nucleotide identity (haplotypes GQ472027 and MK005361) [37], though the similarity of the Mariental virus host *M. namaquensis* sequence from Namibia was lower (95.3%) [11]. Additionally, the cytochrome B sequences of UP12472 shared 93–94.8% (AB731581, LM994810, and LM994813) to 99.9% (KF184310) nucleotide identity to *M. (N.) minutoides* from Zambia, Tanzania, South Africa, and Botswana, respectively. The morphological species identities and the cytochrome B barcodes were in agreement for both host species.

## 4. Discussion

Urban expansion, habitat loss, and climate-driven landscape changes are major drivers of the increase in opportunities for novel zoonotic pathogens to emerge [38,39]. Arenaviruses with both known and unknown zoonotic potential have been reported from African small mammal species. Improving the characterization of such viruses and our ecological understanding of this viral diversity would strengthen our preparedness for potential future viral outbreaks and aid in developing tools to help investigate other related highly pathogenic viruses [3,40].

We identified two arenaviruses from two different rodent species during two years of sampling from the Meletse region in Limpopo. The percentage positivity for arenaviruses was low overall (<1%), with higher rates of detection per genus (nearly 4 to 11%). Of the 13 genera investigated, arenaviruses were only detected from the *Mus* (*Nannomys*) and *Micaelamys* genera. Though *Mastomys* spp. are known hosts of MOPV, LASV, GAIV, LUNV, and Merino-Walk viruses, none of the 73 fecal or 57 kidney samples (representing 23% of all samples tested) were shown positive for arenavirus RNA. This is in contrast with previous reports, which found *Mastomys* as the most regular host for Old World arenaviruses [8,41]. Surveillance of this species at the site will be continued as the sample sizes of *Mastomys* sp. collected may have been too few per season to identify low prevalence mammarenaviruses present in the populations.

UP12291/M. namaquensis/LP/RSA/2017 and UP12472/M. minutoides/LP/RSA/2017 genomes share the closest genetic similarity to Mariental and Lunk viruses, respectively, which were initially reported from other southern African countries [10,11]. These thus represent the first descriptions of these arenaviruses from rodent species in South Africa. Mariental virus was initially described from the lung tissues of a *Micaelamys namaquensis* rodents captured in 2012 in the city of Mariental in Namibia. UP12291/M. namaquensis/LP/RSA/2017 was identified from the kidney, lung, liver, spleen, urine and fecal of the same host species, though separated geographically by 977 km (Mariental town in Namibia to the Meletse region in South Africa). Detection of viral RNA from both urine and fecal material is suggestive of active excretion of viral particles into the environment. According to the International Union for Conservation of Nature (IUCN) red list, the distribution of *Micaelamys namaquensis*, is throughout Southern Africa (including southern Angola, Western Mozambique, Botswana, Zimbabwe, Namibia, Eswatini and South Africa [42]). Other arenaviruses reported from the host, include Bitu, Okahandja, and Witsand viruses from Angola, Namibia, and South Africa [8,9,11], though group in a basal clade to Mariental and UP12291/M. namaquensis/LP/RSA/2017 (Figure 3 and Appendix A). The former viruses and the Mariental lineage are rather divergent and share only 67–71.8% amino acid similarity among the GP protein and 58–63% among NP proteins.

Lunk virus was described from the kidney of *Mus minutoides* captured in 2010 in Kasama, Zambia. RNA of UP12472/M. minutoides/LP/RSA/2017 was detected from the kidney, liver, lung, and spleen of the same host species, though capture sites are separated by 2622 km (Kasama to the Meletse region). *Mus minutoides* has been recorded by the IUCN red list to be extant within Zambia, Zimbabwe, Eswatini, Western Malawi and Mozambique and along coastal regions of Southern Namibia and South Africa [43]. Lunk and UP12472/M. minutoides/LP/RSA/2017 are grouped in a phylogenetic clade (Figure 3 and Appendix A) comprising another *Mus*-hosted arenavirus, Rat arenavirus 1-YN2013, and well as Dandenong virus from humans and LCMV that is associated with both, and share 78–81% and 77–78% amino acid similarity among NP and GP proteins, respectively. The findings would suggest diverse arenaviruses present among the *Mus* host genera.

The arenaviruses described here meet many, though not all, of the ICTV’s species demarcation criteria to be considered members of either the *Mariental* or *Lunk mammarenavirus* species. Both viruses from UP12291/M. namaquensis/LP/RSA/2017 and UP12472/M. minutoides/LP/RSA/2017 originated from the same host species as Mariental and Lunk viruses, respectively, and in similar geographical regions (South Africa, Namibia and Zambia) as these hosts are broadly distributed throughout southern Africa [11]. The nucleoprotein amino acid similarities are above the minimum required 88% sequence identity shared by species, though in the case of UP12291/M. namaquensis/LP/RSA/2017 and Mariental virus, only 88.8% [44]. However, the nucleotide sequence identities of the S and L segments are below the threshold of 80% and 76%, respectively, for either UP12291/M. namaquensis/LP/RSA/2017 and Mariental virus as well as UP12472/M. minutoides/LP/RSA/2017 and Lunk virus. Thus, whether these newly described arenaviruses represent strains of existing species or new species will be based on the discretion of the ICTV’s Arenaviridae study group.

The zoonotic risk of many of the diverse Old World mammarenaviruses described from several African countries is undetermined. Moreover, limited research has been performed on recently identified arenaviruses, particularly regarding their pathogenicity and capability of infecting people or other animals. Known arenavirus receptors include alpha-dystroglycan and transferrin receptor 1 [35], the former used by known pathogenic Old World mammarenaviruses like LASV, MOBV, LUNV, MOPV, Morogoro, and Dandenong viruses as well as LCMV [36]. Alpha-dystroglycan is a conserved and ubiquitous cell surface protein assisting in the adherence to the extracellular matrix, that may be utilized by arenaviruses to avoid the early endocytic pathway [32]. Specifically, arenaviruses interact with the matrigylcan carbohydrate on alpha-dystroglycan, though the exact molecular interactions for recognition have recently been better described [36]. The GP1 subunit terminal residues are conserved as they serve as a recognition motif in cleaving the glycoprotein precursor into GP1 and GP2 subunits and have been recognized as involved in stabilizing the spike glycoprotein to form the matriglycan binding site [32,36]. The conserved residue is ‘RRLL’ among most Old World alpha-dystroglycan tropic mammarenaviruses, with variations including ‘RRLM ‘and ‘RRLA’ [36] (Appendix A). Inference of GP1 subunit similarities and the RRLL recognition motif may suggest that UP12472/M. minutoides/LP/RSA/2017 (and Lunk virus) could be capable of utilizing the alpha-dystroglycan receptor (similarly to LCMV and Dandenong within the same phylogenetic clade). The consequences of a Leu261 replacement with Ile261 on the structural conformation of a stable matriglycan binding site for either Mariental virus or UP12291/M. namaquensis/LP/RSA/2017 is unknown and a unique variation of the RRLL recognition motif (Appendix A). Whether the viruses described here can utilize the alpha-dystroglycan receptor and potentially infect human cells can be practically confirmed with functional studies assessing zoonotic potential [8]. The scope of clinical disease associated with known pathogenic mammarenavirus infection is worldwide, with a spectrum of disease severity ranging from asymptomatic to fatal hemorrhagic fever. Improved serosurveillance studies in communities at risk of contact with rodent species that are known hosts for Old World arenaviruses could provide more insights into possible exposure events.

## 5. Conclusions

Mammarenaviruses have been described from various rodent species throughout most of Africa. Active surveillance activities describe novel diversity and new variants of known species with unique genetic differences. Here, we described limited detected diversity from a small number of rodents actively infected with arenaviruses, in contrast to previous similar studies. Systematic longitudinal surveillance targeting larger sample sizes of specific species may shed light on possible higher risk periods of arenavirus shedding or potential geographic or intra-species-interactions contributing to arenavirus circulation among rodent populations. Identifying arenaviruses from South Africa (described here and other literature [8]) that may have the potential of causing human disease warrants improved surveillance of rodents at interfaces closer to human settlements to investigate possible ongoing exposure events.

## Figures and Tables

**Figure 1 viruses-15-00099-f001:**
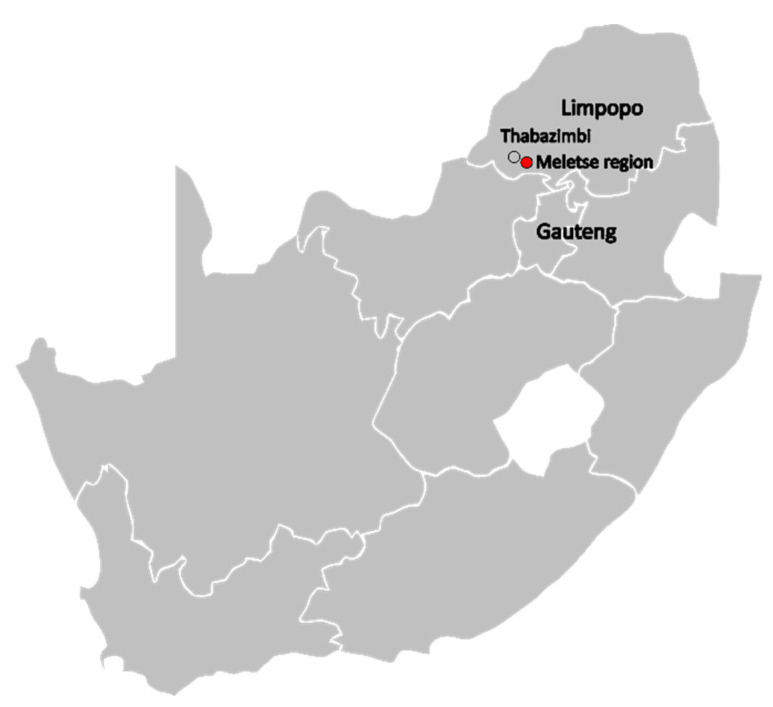
Map of the study site within South Africa, indicating the Meletse region compared to the nearest town (Thabazimbi) relative to the provinces of Limpopo and Gauteng.

**Figure 2 viruses-15-00099-f002:**
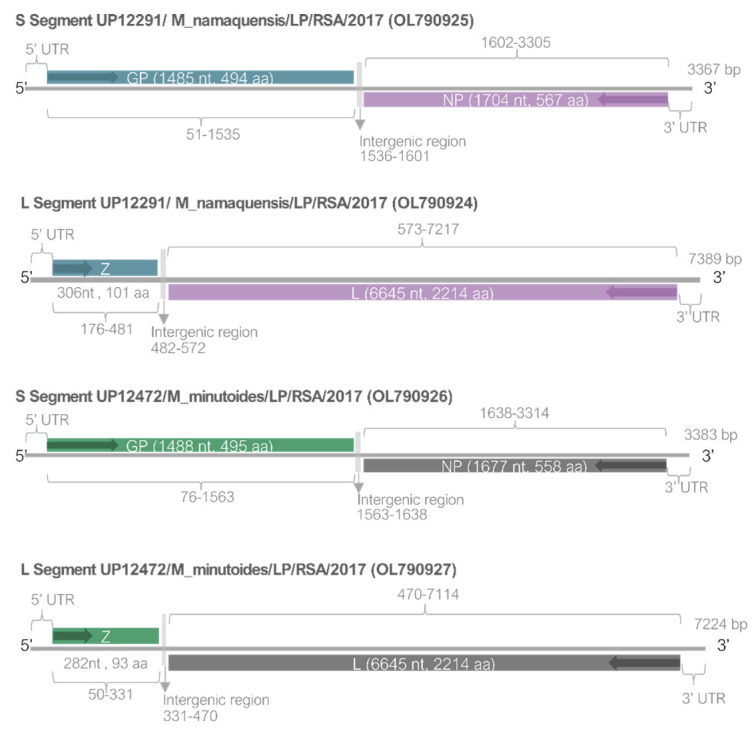
Gene organization of genome segments of arenaviruses detected in Limpopo, South Africa. Positions of open reading frames are given, including intergenic loop regions, sizes of Glycoprotein (GP), Nucleoprotein (NP), Zinc-binding protein (Z), and RNA-directed RNA polymerase (L) genes as nucleotides (nt) and amino acids (aa).

**Figure 3 viruses-15-00099-f003:**
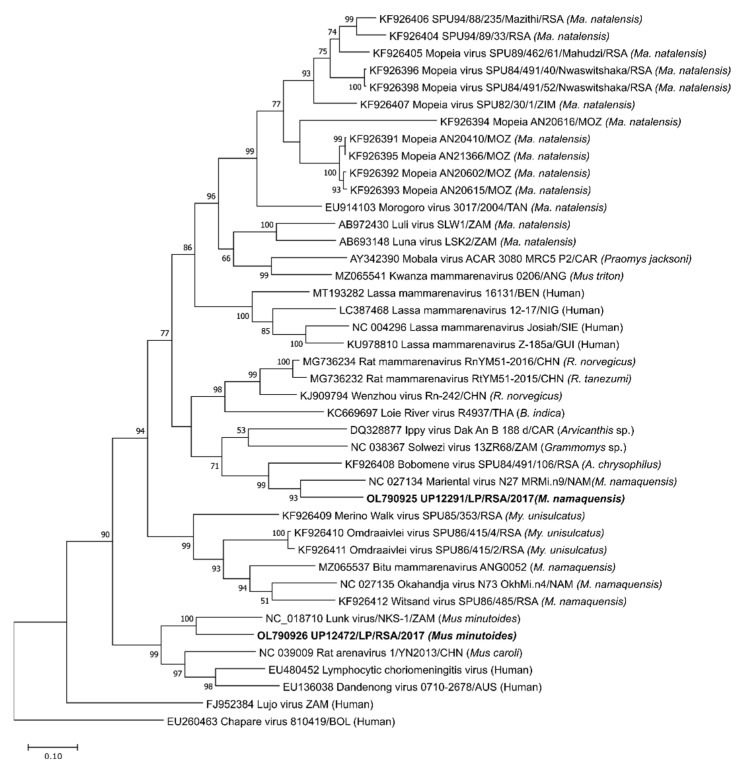
Phylogeny of a 912-nucleotide region of the S segment nucleoprotein gene constructed by maximum likelihood approach using the Jukes-Cantor substitution model in MegaX with 1000 bootstrap replicates. Sequences in bold denote those reported from this study. All host species of origin are indicated, including countries from which viruses were reported as standard 3-letter country codes.

**Table 1 viruses-15-00099-t001:** Total samples from non-volant mammals tested for arenaviruses according to host genus.

Genus	Fecal Samples	Positive Fecal Samples	Kidney Samples	Positive Kidney Samples	Total Samples Tested per Genus
*Acomys* sp.	3	-	2	-	5
*Aethomys* sp.	67	-	47	-	114
*Crocidura* sp.	2	-	2	-	4
*Elephantulus* sp.	4	-	2	-	6
*Gerbilliscus* sp.	75	-	34	-	109
*Graphiurus* sp.	2	-	2	-	4
*Lemniscomys* sp.	7	-	9	-	16
*Mastomys* sp.	70	-	55	-	125
*Micaelamys* sp.	12	1 (8.33%)	10	1 (10%)	22
*Mus (Nannomys)* sp.	25	-	27	1 (3.7%)	52
*Saccostomus* sp.	22	-	19	-	41
*Steatomys* sp.	28	-	33	-	61
*Suncus* sp.	1	-	3	-	4
Overall Total	318	1 (0.31%) ^#^	245	2 (0.82%) *	563

^#^ a 95% confidence interval of 0.01–1.73%. * a 95% confidence interval of 0.1–2.9%.

**Table 2 viruses-15-00099-t002:** Similarities between reference species and genomes identified in this study.

Gene or Segment Compared	Similarity between UP12291 & Mariental Virus (nt/aa)	Similarity between UP12472 & Lunk Virus (nt/aa)
S segment (full segment)	77.1% nt	76.9% nt
Glycoprotein (GP)	75.2% nt/86.7% aa	76.5% nt/89.6% aa
Nucleoprotein (NP)	78.6% nt/88.8% aa	79.7% nt/94.2% aa
L segment (full segment)	65.6% nt (* 74% nt)	71.3% nt
Zinc-binding protein (ZP)	70.5% nt/72.5% aa	66.3% nt/60.6% aa
RNA-directed RNA polymerase (L) gene	67.9% nt/67.2% aa * (73.9% 75.7% aa)	72.7% nt/76.4% aa ^#^ (72.4% nt/77.3% aa)
GP1/GP2 cleavage motif	RRIL	RRLM

* Removal of nearly 870 bp region (possible Mariental artefact) resulted in a higher % identity of the segment/ L gene. ^#^ Similarity between only the shorter ORF of Lunk’s L gene with the corresponding region on UP12472.

## Data Availability

Accession numbers of genetic sequence data are provided in the manuscript, with supporting information available under Appendix A.

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
