# Peer review of "Host-Associated Distribution of Two Novel Mammarenaviruses in Rodents from Southern Africa"

_viruses, 2022, doi:10.3390/v15010099_

Round 1

Reviewer 1 Report

This report shows that the authors have identified two novel mammarenaviruses that are closely related to Mariental and Lunk viruses in rodents from Limpopo province, South Africa. The manuscript is well-written. The methodology is also described in understandable detail. There are few issues that need to be amended before acceptance.

1. The title does not faithfully indicate what the authors studied. Please describe in title that the authors found two novel mammarenaviruses from rodents in South Africa.

2. Although authors show the results of surveillance for hantaviruses in the supplementary files, there is no description regarding this hantavirus study in the main text.

3. “RNA-directed RNA polymerase” should be “RNA-dependent RNA polymerase”.

Author Response

Reviewer 1:

Comments and Suggestions for Authors

This report shows that the authors have identified two novel mammarenaviruses that are closely related to Mariental and Lunk viruses in rodents from Limpopo province, South Africa. The manuscript is well-written. The methodology is also described in understandable detail. There are few issues that need to be amended before acceptance.

We thank the reviewer for a very positive response to the manuscript. We have incorporated the reviewer’s suggestions:

  1. The title does not faithfully indicate what the authors studied. Please describe in title that the authors found two novel mammarenaviruses from rodents in South Africa:

- title changed to: Host-associated distribution of two novel mammarenaviruses in rodents from Southern Africa

  1. Although authors show the results of surveillance for hantaviruses in the supplementary files, there is no description regarding this hantavirus study in the main text.

- The hantavirus surveillance are negative results, and instead of leaving it out of the manuscript we felt it was relevant to still report this surveillance (even though no hantavirus nucleic acids were identified). We didn’t mention this in the manuscript as we did not wish to add confusion or retract from the arenavirus surveillance. We have added an additional line to the methods section for clarity directing the reader to the supplementary file to review the hantavirus surveillance results.

 “Hantavirus surveillance was also conducted according to Klempa [26], though no nucleic acids were identified (see supplementary S1 and S2).”

  1. “RNA-directed RNA polymerase” should be “RNA-dependent RNA polymerase”.

Thank you for highlighting this. We used the convention utilized by the ICTV where they use the terminology of “RNA-directed RNA polymerase” when referring to the L segment gene.

Reviewer 2 Report

this is a nice work describing two mamarenaviruses in South Africa. This is the first time finding these viruses in this country, and it is the most important finding from this work.

I found the MS clear and well written. I have just some minor suggestions to the authors.

Introduction:

- I would move line 62-68 from page 2 to the discussion section. It seems to me they are more suitable for the discussion part when discussing discrepancies with ICTV procedures.

Methods:

- some additional details regarding NGS procedures would be appreciated, both in term of sequecing informations and bioinfamrtic tools

Results:

- the classic plot for PASC analysis could have been provided to help the reader understand similarities

- Some sentences regarding the geographic distribution of the 2 host species could have been provided. Maybe also a figure could help.

Discussion

- The authors could have provided some hypotheses /speculations why no Mastomys was found positive to mammarenavirus RNA

Author Response

Reviewer 2:

Comments and Suggestions for Authors

this is a nice work describing two mamarenaviruses in South Africa. This is the first time finding these viruses in this country, and it is the most important finding from this work.

“Thank you to the reviewer for a positive response to the submitted manuscript.”

I found the MS clear and well written. I have just some minor suggestions to the authors.

Introduction:

- I would move line 62-68 from page 2 to the discussion section. It seems to me they are more suitable for the discussion part when discussing discrepancies with ICTV procedures.

“We thank the reviewer for the suggestion. We have moved this section around in the manuscript prior to submission and ultimately feel that the description of species demarcation at the start of the manuscript puts the results into better perspective once the reader is reminded of the guidelines at the start of the manuscript – particularly once we begin reporting on the similarities between relevant genes and segments in the results section. We then discuss these guidelines again in the discussion section.”

Methods:

- some additional details regarding NGS procedures would be appreciated, both in term of sequecing informations and bioinfamrtic tools

“We thank the reviewer for the comment. There is not much more detail that we can provide on the sequencing preparation as it was straight-forward amplicon sequencing with the PCRs used explained in detail, though we have added some additional text regarding the pooling, and we also added more detail on the bioinformatics. We also wish to highlight that we validated the bioinformatics assembly with follow up PCR and sanger sequencing.”

Added the following:

“Amplified products were gel purified and concentration determined with Qubit High Sensitivity double-stranded DNA assay (ThermoScientific, USA). Amplicon regions were combined in equal proportions per segment per virus requiring four libraries. Sequencing was done at the National Institute for Communicable Diseases Sequencing Core Facility (Sandringham, SA). After library prep with the Nextera XT Library prep kit (Illumina, USA), the amplicons were sequenced at 10 million reads. CLC Genomics Workbench v6 was used to quality trim raw reads, and paired-end reads were used to perform de novo assemblies to recover full segments. NCBI’s ORF finder (online tool available at https://www.ncbi.nlm.nih.gov/orffinder/) was utilized to validate open reading frames assembled for correct genes and functional proteins. Sequences were submitted to Genbank with accession numbers: OL790924-OL79092. Following segment assembly with de novo assembly, the sequences toward the 5’ and 3’ ends of the segments and regions of significant difference in comparison to reference genomes were confirmed by re-amplification with sequence-specific primers (using kidney and lung RNA extracts) and Sanger sequencing. The primer sequences and binding locations are listed in Table S2.”

Results:

- the classic plot for PASC analysis could have been provided to help the reader understand similarities

“these plots have been added to the supplementary file as figure S4-S7”

- Some sentences regarding the geographic distribution of the 2 host species could have been provided. Maybe also a figure could help.

“ thank you for the suggestion. As these species are commonly encountered a text description seems suitable. We have added the following to the discussion:

“According to the IUCN red list, the distribution of Micaelamys namaquensis, is throughout Southern Africa (including southern Angola, Western Mozambique, Botswana, Zimbabwe, Namibia and South Africa [42]).” And “Mus minutoides has been recorded by the IUCN red list to be extant within Zambia, Zimbabwe, Eswatini, Western Malawi and Mozambique and along coastal regions of Southern Namibia and South Africa [43].

Discussion

- The authors could have provided some hypotheses /speculations why no Mastomys was found positive to mammarenavirus RNA

“We have added the following: Surveillance of this species at the site will be continued as the sample sizes of Mastomys sp. collected may have been too few per season to identify low prevalence mammarenaviruses present in the populations.“

We also edited the manuscript's syntax and grammar via 'Grammarly'